# Stepwise Method and Factor Scoring in Multiple Regression Analysis of Cashmere Production in Liaoning Cashmere Goats

**DOI:** 10.3390/ani12151886

**Published:** 2022-07-23

**Authors:** Yang Meng, Boqi Zhang, Zhiyun Qin, Yang Chen, Xuesong Shan, Limin Sun, Huaizhi Jiang

**Affiliations:** 1College of Animal Science and Technology, Jilin Agricultural University, Changchun 130118, China; my17649988337@126.com (Y.M.); qinzhiyunl@126.com (Z.Q.); chenyang7419@163.com (Y.C.); shanxuesong@jlau.edu.cn (X.S.); sheepandgoatjlau@sina.com (L.S.); 2College of Animal Sciences, Jilin University, Changchun 130062, China; zhangbq19@mails.jlu.edu.cn

**Keywords:** regression analysis, stepwise analysis, factor analysis, cashmere yield, Liaoning cashmere goats, body weight measurements

## Abstract

**Simple Summary:**

Liaoning cashmere goat is a world-famous cashmere goat breed known for its stable genetic performance and high cashmere yield. This study investigated the relationship between certain body measurements, body weight, and cashmere yield in Liaoning cashmere goats using stepwise and factor score analyses in a multiple regression analysis. These results provide a standardized regression equation and reliable information on multiple trait considerations that should be taken to improve cashmere yield. To clarify the relationship and influence among the body circumference, body weight, and cashmere yield of Liaoning cashmere goat, aiming to provide the basis for the breed selection and germplasm identification of Liaoning cashmere goat.

**Abstract:**

Liaoning cashmere goat is a well-known local cashmere goat breed in China and even in the world. It is famous for producing cashmere with superior quality and high yield. Cashmere yield, body measurements, and body weight are the primary indicators of cashmere goat breeding, but the correlation between them is not yet clear. Therefore, this study investigated the relationship between certain body measurements, body weight, and cashmere yield in Liaoning cashmere goats using stepwise and factor score analyses in a multiple regression analysis. For this purpose, the body measurements (body slanting length (BSL), body height (BH), chest circumference (CC), pipe circumference (PC), chest depth (CD), chest width (CW), hip breadth (HB), body weight (BW) and cashmere yield (CY)) of 200 (2-year-old) Liaoning cashmere goats were collected. Stepwise analysis of the results showed that body weight had the greatest direct effect on cashmere yield, followed by hip breadth, while chest circumference mainly affected cashmere yield indirectly. The results of factor score analysis showed that the independent variable can be represented by two factors, which explained 49.596% and 12.095% of the total variance, respectively. The factor scores used in the regression analysis explained 75.8% of the total variance in Liaoning cashmere yield. The above studies show that the growth traits of Liaoning cashmere goats are closely related to the cashmere yield. Growth traits should be considered important factors in breed selection, germplasm identification, and rearing.

## 1. Introduction

In animal husbandry, body weight and cashmere yield are important indicators that directly affect the economic benefits provided by cashmere goats [1,2]. Body measurements can directly reflect the size of the animal, structure, and development of its body. It also indirectly reflects the development of tissues and organs and is closely related to the physiological function, production performance, disease resistance, and adaptability of livestock to external living conditions. 

Liaoning cashmere goat is an indigenous breed in China [3], living in temperate humid climate regions, mainly for the production of cashmere [4]. It is famous for high cashmere yield and long and thin cashmere fibers. Moreover, it is the cashmere goat breed with the highest average cashmere yield in China and even in the world [5]. The cashmere produced by Liaoning cashmere goat is a natural fiber with excellent texture. Its fabric is not only light and strong but also has a comfortable wearing performance. Like other wool-producing sheep breeds, the double coat of the Liaoning cashmere goat comes from the primary hair follicles and the secondary hair follicles with the characteristics of self-renewal and periodic growth in their skin, respectively [6]. This makes the coat of the breed have a physiological phenomenon of alternating growth and shedding. The tissue structure and development process of the Liaoning cashmere goat are closely related to the yield and quality of cashmere. The growth and development of cashmere goats, the level of cashmere yield, and the quality of cashmere all determine the economic benefits of cashmere goat breeding. Therefore, its growth and development, cashmere yield and cashmere quality, and other performance indicators have become the main economic traits of cashmere goat breeding and also become the breeding target of cashmere goat breed selection. In this study, the relationship between other body measurements to body weight and cashmere yield is very important for the breed selection and germplasm identification of Liaoning cashmere goats.

Multiple regression models are the most commonly used predictive models to explain the relationship between body measurements, body weight, and cashmere yield [7,8,9]. However, this method has certain limitations. Multicollinearity can occur between independent variables, leading to inaccurate statistical interpretations of the coefficients of the regression parameters [10,11]. Therefore, it was necessary to test for multicollinearity. One way to avoid such problems is to use factor analysis to obtain independent factor scores [12,13].

Factor analysis refers to the study of statistical techniques to extract common factors from groups of variables. It is a method of explaining the structure and is very important in the application of multivariate statistical methods [14]. New variables (factors) can be extracted from multiple difficult-to-interpret and correlated variables with minimal loss, and multicollinearity among the original independent variables can be eliminated [15]. To date, research using stepwise analysis and factor scoring in multiple regression analysis to predict cashmere yield in Liaoning cashmere goats is very limited.

Therefore, in this study, stepwise analysis and factor scores were used in the multiple regression method to analyze the relationship between body measurements, body weight, and cashmere yield in Liaoning cashmere goats. 

## 2. Materials and Methods

### 2.1. Animals 

The research materials were taken from the nucleus herd of cashmere goat breeding farms in Benxi City, Liaoning Province (East longitude 123°34′~125°46′, north latitude 40°49′~41°35′, the annual average temperature is 4–15 °C), and 2-year-old adult ewes were selected for body measurement, body weight measurement and wool (cashmere) sample collection. The 200 goats selected for the experiment had complete pedigree records, and their body condition, appetite, and health were all good. After weaning at 90 days of age, the diet was designed according to the NRC [16] standard. The experimental group was reared in the whole house, and each goat was fed 2 kg/d of alfalfa hay and 0.5 kg/d of concentrated feed. Concentrated feed is self-made mixed feed (crude protein content 14%). The feeding conditions were quiet, ventilated, clean, and dry; the temperature was suitable, and feed and drinking water were added regularly. After the measurement and collection, the standard feeding was continued in the experimental site without the use of anesthesia and other drugs throughout the process.

### 2.2. Experimental Design

The cashmere yield, body measurements, and body weight of Liaoning cashmere goats are important indicators that affect the economic benefits. In this experiment, 200 Liaoning cashmere goats were selected, and the phenotypic values of body slanting length (BSL), body height (BH), chest circumference (CC), pipe circumference (PC), chest depth (CD), chest width (CW), hip breadth (HB), body weight (BW), and cashmere yield (CY) were measured. According to the measured phenotypic values, the correlation between each trait and cashmere yield was analyzed by stepwise analysis and factor score analysis in multiple regression analysis. The regression equation of the estimated cashmere yield was obtained by analysis. The purpose is to provide the basis for the breeding process and germplasm identification of the Liaoning cashmere goat.

As the sample size in this study was not large enough (200), this first result can serve as a basis for further studies with large sample size and longer follow-up period.

### 2.3. Methods and Measurements

The cashmere yield (Y), body slanting length (X_1_), body height (X_2_), chest circumference (X_3_), pipe circumference (X_4_), chest depth (X_5_), chest width (X_6_), hip breadth (X_7_), and body weight (X_8_) of 200 Liaoning cashmere goats were measured using a leather ruler, iron ruler, measuring stick, and an electronic scale.

The multiple regression equation describes the relationship between a dependent variable Y and k independent variables (X_1_, X_2_, …, X_k_). Assuming that the influence of each independent variable on the dependent variable Y is linear, then the mean value of Y changes uniformly with the change in the independent variable $X_{i}$ when other independent variables remain unchanged; Equation (1) is called the population regression model.
(1)$$Y=\beta_{0}+\beta_{1}X_{1}+\beta_{2}X_{2}+\ldots +\beta_{k}X_{k}+\epsilon .$$

In the equation, β_0_, β_1_, β_2_,…, β_k_ are regression parameters, and ϵ is an unmeasurable random error.

The basic tasks of regression analysis are to use the sample data to estimate the model parameters and hypothesis testing on model parameters.

A regression model is applied to predict the dependent (explained) variable. Equation (2) is the population regression equation or population regression function.
(2)$$E(Y|X_{1},X_{2},\ldots ,X_{k})=\beta_{0}+\beta_{1}X_{1}+\beta_{2}X_{2}+\ldots +\beta_{k}X_{k}.$$

In practical problems, the overall parameters β_0_, β_1_, β_2_,…, β_k_ are often unknown, and the corresponding estimated values $\hat{\beta_{0}}$,$\hat{\beta_{1}}$,$\hat{\beta_{2}}$, …, $\hat{\beta_{k}}$ are sample observations. Equation (3) is the sample regression equation or sample regression function. $\hat{Y}$ is the point estimate of $E(Y|X_{1},X_{2},\ldots ,X_{k})$.
(3)$$\hat{Y}=\hat{\beta_{0}}+\hat{\beta_{1}}X_{1}+\hat{\beta_{2}}X_{2}+\ldots +\hat{\beta_{k}}X_{k}.$$

There are two basic steps in the process of selecting variables by stepwise regression: (1) to remove variables that are not significant after testing from the regression model and (2) to introduce new variables into the regression model. Commonly used stepwise element selection methods are the forward and backward regression.

Forward regression establishes a univariate regression equation, as in Equation (4).
(4)$$Y=\beta_{0}+\beta_{i}X_{i}+\epsilon ,i=1,\ldots ,p$$

The variable $X_{i}$, the value of the regression coefficient F test statistic is $F_{1}^{\left(1\right)}$,…, $F_{p}^{\left(1\right)}$, and the maximum value $F_{i_{1}}^{\left(1\right)}$ are calculated as visible from Equation (5).
(5)$$F_{i_{1}}^{\left(1\right)}=max\{F_{1}^{\left(1\right)},\ldots ,F_{p}^{\left(1\right)}\}$$

For a given significance level α, the corresponding critical value was recorded as $F^{\left(1\right)}$, $F_{i_{1}}^{\left(1\right)}\geq F^{\left(1\right)}$, then $X_{i_{1}}$ is introduced into the regression model, and $I_{1}$ is recorded as the set of selected variable indicators.

A binary regression model is built, and the statistical value of the regression coefficient F test of the variable is calculated as $F_{k}^{\left(2\right)}\;\left(k\notin I_{1}\right)$, selected for its maximum value $F_{i_{2}}^{\left(2\right)}$, and the corresponding independent variable is marked as $i_{2}$, as shown in Equation (6).
(6)$$F_{i_{2}}^{\left(2\right)}=max\{F_{1}^{\left(2\right)},\ldots ,F_{i_{1}-1}^{\left(2\right)},F_{i_{1}+1}^{\left(2\right)},\ldots ,F_{p}^{\left(2\right)}$$

For a given significance level α, the corresponding critical value is $F^{\left(2\right)}$, and $F_{i_{2}}^{\left(1\right)}\geq F^{\left(2\right)}$ is variable $X_{i_{2}}$ introduced into the regression model.

This method is repeated every time one of the independent variables that are not introduced into the regression model is selected until no variables are introduced.

The backward regression is the opposite of the forward one. Initially, the regression equation, including all independent variables, was fitted, and the hypothesis test criteria for the independent variables that remained in the regression equation and were not excluded were pre-specified. In the entire process, only the independent variables were eliminated. Once the independent variables were eliminated, the regression equation was no longer considered.

Multicollinearity may occur between the independent variables that are used to predict the results of the multiple regression analysis. As a result, the influence of the independent variable on the dependent variable intensifies, and the variance and covariance of the regression coefficients increase. Regardless of the R^2^ value, only a small number of independent variables were significant according to the *t*-test [10,11].

Generally, multicollinearity is diagnosed by the variance inflation factor (VIF) and tolerance, and VIF and tolerance are reciprocal of each other, with each independent variable having its VIF and tolerance. Therefore, it is necessary to calculate the tolerance given in Equation (7) and VIF given in Equation (8) to reveal multicollinearity.
(7)$$\mathrm{tolerance}=1-R_{i}^{2}$$
(8)$$\mathrm{VIF}=\frac{1}{1-R_{i}^{2}}$$

Generally, when VIF > 10, it indicates that there is a multicollinearity problem [17]. An estimation method for the factor scores can be used to eliminate multicollinearity issues between the dependent variables. Factor analysis can explain the putative relationships between multiple variables and can represent multiple variables with fewer factors [15]. Z can be used to represent the px1-dimensional random vector, λ represents the pxm-dimensional self-loading matrix, F represents the mx1-dimensional factor vector, and ϵ represents the px1-dimensional error vector [18,19] to establish the matrix form factor Equation (9).
(9)Z=λF+ϵ.

In factor analysis, the Kaiser–Meyer–Olkin (KMO) and Bartlett were used to test the separability of the correlation matrix to the factors [20]. Obtained by the KMO test greater than the threshold of 0.5, indicating that the relationship between variables can be explained by other variables [21]. Considering factor loadings, a varimax rotation was used, and factor coefficients were used to obtain the selected factor scores [13,22]. Multicollinearity was resolved using these coefficients through the factor scores. The number of factors used in the multiple regression model must be represented by eigenvalues greater than one obtained from the correlation matrix [15,20].

### 2.4. Statistical Analysis

In this study, all statistical calculations for Liaoning Cashmere goats were performed using SPSS 26.0.

## 3. Results

Table 1 shows the descriptive statistics for body measurements, body weight, and cashmere yield of 200 Liaoning cashmere goats at 2 years of age. Statistical analysis showed that the coefficients of variation of body slanting length and chest circumference were the smallest at 1.63% and 1.67%, respectively; the chest depth and width were similar at 5.17% and 5.16%, respectively; and the maximum coefficient of variation of hip breadth was 6.76%. It can be seen that the heritability among the traits is relatively stable. 

At the same time, Appendix A shows the F test statistic of each trait. The significance of the F test statistic is to evaluate how well the sample results can represent the true degree of the population. The larger the F value, the greater the probability of passing the test, the smaller the residual, and the higher the accuracy of the simulation. 

Table 2 shows the Pearson correlation coefficient and significance test for the body measurements, body weight, and cashmere yield of Liaoning cashmere goats. Table 2 shows that all traits and cashmere yield are highly significantly positively correlated (*p* < 0.01), of which, the correlation coefficient between body weight and cashmere yield was the largest (r = 0.734; *p* < 0.01), followed by hip breadth (r = 0.681; *p* < 0.01), chest circumference (r = 0.633; *p* < 0.01), chest width (r = 0.571; *p* < 0.01), and body height (r = 0.326; *p* < 0.01). The phenotypic correlation coefficient between body height and body slanting length showed a significant positive correlation (0.01 < *p* < 0.05). There was no significant correlation between chest depth and body height (*p* > 0.05); however, body weight and height were significantly positively correlated (*p* < 0.05). There were significant positive correlations among the other traits (*p* < 0.01). 

In path coefficient analysis, the degree of direct and indirect influence between the indicators can be shown, while the correlation coefficient can only reflect the phenotypic correlation between the two traits. Table 3 shows the results of the path coefficient analysis between the body measurements, body weight, and cashmere yield of Liaoning cashmere goats. It can be seen from the table that the direct influence of body weight (0.433) on cashmere yield is the greatest, followed by hip breadth (0.248). However, body slanting length, body height, chest circumference, pipe circumference, chest depth, and chest width had little direct influence on cashmere yield, among which Pipe circumference (0.064) was the smallest. Chest circumference (0.559) had the greatest indirect effect on cashmere yield, followed by chest width (0.493), pipe circumference (0.449), and hip breadth (0.432), while the indirect effect of body height (0.239) was the smallest. In summary, body weight affected cashmere yield through direct action, while chest circumference and chest width indirectly affected production. To sum up, for the breeding of Liaoning cashmere goats, while considering increasing cashmere yield, traits such as body weight, hip breadth, and chest circumference should be taken into account.

The regression analysis was carried out by stepwise method, and the regression equation of body measurements, body weight, and cashmere yield of Liaoning cashmere goat was obtained.

The linear regression equation obtained was: Y = −386.562 + 2.899X_1_ + 1.781X_2_ + 1.83X_3_ + 25.707X_4_ + 2.889X_5_ + 2.836X_6_ + 10.29X_7_ + 5.443X_8_,

The standardized regression equation was:Y = 0.071X_1_ + 0.088X_2_ + 0.074X_3_ + 0.064X_4_ + 0.087X_5_ + 0.079X_6_ + 0.248X_7_ + 0.433X_8_.

In the standardized regression equation, the regression relationship between hip breadth, body weight, and cashmere yield was extremely significant (*p* < 0.01), and the regression relationship between body height and cashmere yield was significant (0.01 < *p* < 0.05). This suggested a relationship between these traits and cashmere yield.

Multiple regression analysis is used to analyze a dataset to prevent the reliability of the calculation results from being affected by the high correlation between the variables. To analyze the multicollinearity problem, the expected coefficient, standard error, test statistic, tolerance, and VIF value of each parameter in the multiple regression analysis were investigated, as shown in Table 4. The tolerance of all traits was greater than 0.1, and the VIF value was less than 10. This proved that there was no multicollinearity problem in multiple regression analysis. 

Factor score analysis is mostly used when the independent variables have multicollinearity, which can reduce the risk of inaccurate interpretation of the parameters in the model. Kaiser–Meyer–Olkin (KMO) and Bartlett’s test are prerequisites for factor analysis; in this research, we considered the KMO value (0.865) and the Bartlett test value (*p* < 0.01) to test the separability of the correlation matrix decomposition into factors and determine whether the data can be subjected to factor analysis [20].

The results of factor analysis are presented in Table 5. The selected eigenvalues in the results are all greater than 1 and can be used as independent variables in the regression analysis [10,13,22]. The two selected factors explained 49.596% and 12.095% of the total variance of all variables, respectively, adding up to a total of 61.691%. The factor loads given in Table 5 show the relationship between the investigated independent variables and factors. It also shows the relationship between the factor loads of the independent variables and factors. The bold values in the table show the highest correlation between the investigated traits and factors. As a result of the analysis, factor loads of the independent variables for the first factor were body slanting length (0.654), chest circumference (0.684), pipe circumference (0.716), and chest depth (0.793). In addition, body height (0.745), chest width (0.723), and hip breadth (0.732) were determined as the second factors.

Moreover, the factor score coefficients of the two factors obtained by factor analysis were used as independent variables for Liaoning cashmere goats, and the results are shown in Table 6 to determine the significant factors for predicting cashmere yield in Liaoning cashmere goats.

As shown in Table 6, the regression analysis results of the factor scores showed that each factor was statistically significant (*p* < 0.001) as an independent variable. With the use of factor scores in the model, the multicollinearity problem was solved, and VIF = 1 was found. The factor scores used in the model explained 75.8% of the total variance in cashmere yield of Liaoning cashmere goats. After these results, the cashmere yield estimation equation can be established, and it is expressed as: CY = 18.816 Factor1 + 21.730 Factor2

It is expected that Liaoning cashmere goats with higher body slanting length, chest circumference, pipe circumference, chest depth, body height, chest width, and hip breadth values have higher cashmere yield because of similar signs of rotated factor loads and regression coefficients of factor scores. Moreover, the cashmere yield was positive in factor1 and factor2.

## 4. Discussion

Cashmere yield and growth traits are important indicators to measure the economic benefits of cashmere goats, and they are also the primary indicators of breeding, both of which are affected by genetics, nutrition, and environment. The path coefficient analysis can show the degree of direct and indirect influence between the indicators, while the correlation coefficient can only reflect the phenotypic correlation between the two traits. Therefore, to explore the correlation between cashmere yield and growth traits of Liaoning cashmere goats, the breeding goal of indirectly increasing cashmere yield can be achieved by selecting the performance values of the corresponding traits that are easy to measure. In the breeding practice of domestic animals such as lambs [23,24], sheep [18] and goats [7,25], cattle [19], poultry [20,26], and aquaculture animals [10,27], countries use the correlation regression analysis method and correlation coefficient analysis between the traits to explore the phenotypic correlation and the direct relationship between the target traits, and the analysis of the degree of indirect influence accelerates the breeding process of the corresponding animals. At the same time, it is indicated that cashmere goat breeding should not only consider cashmere yield and growth traits but also need to consider the effects of other factors such as nutrient levels and feeding management on production performance.

In this study, a stepwise regression analysis was performed, and statistical data were used to investigate tolerance and VIF values, demonstrating the absence of multicollinearity (VIF < 10) among the variables. However, many studies with multicollinearity used factor score analysis to solve the problem [22,23,24,28]. Therefore, factor analysis was used in this study. The obtained R^2^ and adjusted R^2^ values were 70.7% and 69.4%, respectively, which were lower than those obtained in previous studies [24,28,29]. To verify the separability of the correlation matrix to the factors, the Bartlett test value (798.31; *p* < 0.01) and a KMO value of 0.865 were determined, which were similar to the results obtained in other studies [18,22].

In the factor analysis of this study, two factors were used as independent variables in the regression analysis, which explained 49.596% and 12.095% of the total variance, respectively, with a total of 61.691%. Similar variables with a three-factor [13] and five-factor structures [22] were also used in the regression analysis.

Similar variables with a two-factor structure were used as independent variables to predict cashmere production in Liaoning cashmere goats, and the effects of all the factors were determined to be significant (*p* < 0.001). The factor scores used for regression analysis explained 75.8% of the total variance of the two Liaoning cashmere goats. Yalcin Tahtali [24] determined 75.4% of the total variance of Romano lambs. Cankaya et al. [22] determined 73.1% of the total variance of Karayaka lambs, a similar trend to our results.

## 5. Conclusions

The results of the stepwise analysis showed that there was no multicollinearity between body measurements, body weight, and cashmere yield, and body slanting length, body height, chest circumference, pipe circumference, chest depth, chest width, hip breadth, body weight, and cashmere yield were all significantly positively correlated (*p* < 0.01). The direct effect of body weight on cashmere yield was the greatest, and the indirect effect of chest circumference was the greatest. The standardized regression equation for estimating the cashmere yield of Liaoning cashmere goats was:Y = 0.071X_1_ + 0.088X_2_ + 0.074X_3_ + 0.064X_4_ + 0.087X_5_ + 0.079X_6_ + 0.248X_7_ + 0.433X_8_.

The results of the factor score analysis showed that body measurements and weight can be represented by two factors. These two factors explained 49.596% and 12.095% of the total variance, respectively, thus adding up to 61.691%. As an independent variable, each factor was statistically significant (*p* < 0.001). The factor scores used in the model explained 75.8% of the total variance in cashmere yield in Liaoning cashmere goats.

This study proposes that in the future breeding process, while considering the high production of cashmere, it is necessary to take into account the traits of body weight, hip breadth, chest circumference, body height, etc. The results of this study elucidated the relationship and effect between body measurements, body weight, and cashmere yield in Liaoning cashmere goats and aimed to provide a basis for breed selection and germplasm identification of Liaoning cashmere goats.

This study still has limitations. Since the sample size in this study was not large enough (200), this first result can serve as a basis for further studies with a large sample size and longer follow-up period.

## Figures and Tables

**Table 1 animals-12-01886-t001:** Descriptive statistics of certain body measurements, body weight, and cashmere yield of Liaoning cashmere goats.

Parameter	n	Mean	Standard Deviation (SD)	Coefficient of Variation (CV) %
Body slanting length (BSL)	200	49.51	0.81	1.63
Body height (BH)	200	42.46	1.62	3.82
Chest circumference (CC)	200	79.88	1.33	1.67
Pipe circumference (PC)	200	4.25	0.08	1.94
Chest depth (CD)	200	19.12	0.99	5.17
Chest width (CW)	200	17.8	0.92	5.16
Hip breadth (HB)	200	11.74	0.79	6.76
Body weight (BW)	200	46.26	2.62	5.66
Cashmere yield (CY)	200	566.37	32.93	5.81

**Table 2 animals-12-01886-t002:** Pearson correlation analysis of body measurements between body weight and cashmere yield of Liaoning cashmere goats.

Traits	CY (Y)	BSL (X_1_)	BH (X_2_)	CC (X_3_)	PC (X_4_)	CD (X_5_)	CW (X_6_)	HB (X_7_)	BW (X_8_)
CY (Y)	—								
BSL (X_1_)	0.413 **	—							
BH (X_2_)	0.326 **	0.148 *	—						
CC (X_3_)	0.633 **	0.488 **	0.338 **	—					
PC (X_4_)	0.512 **	0.352 **	0.219 **	0.658 **	—				
CD (X_5_)	0.427 **	0.340 **	0.041	0.454 **	0.491 **	—			
CW (X_6_)	0.571 **	0.254 **	0.319 **	0.460 **	0.338 **	0.309 **	—		
HB (X_7_)	0.681 **	0.299 **	0.343 **	0.581 **	0.486 **	0.326 **	0.628 **	—	
BW (X_8_)	0.734 **	0.339 **	0.173 *	0.537 **	0.382 **	0.328 **	0.480 **	0.529 **	—

Note: ** indicates an extremely significant correlation (*p* < 0.01); * indicates a significant correlation (*p* < 0.05); no asterisk indicates no significant correlation (*p* > 0.05). There is no comparative significance between the same data, ^—^ used to fill the gap.

**Table 3 animals-12-01886-t003:** Path coefficient analysis between body measurements, body weight, and cashmere yield of Liaoning cashmere goats.

Traits	Correlation Coefficient	Direct Impact	Indirect Effects								
			Sum	BSL (X_1_)	BH (X_2_)	CC (X_3_)	PC (X_4_)	CD (X_5_)	CW (X_6_)	HB (X_7_)	BW (X_8_)
BSL (X_1_)	0.413	0.071	0.343		0.013	0.036	0.023	0.03	0.02	0.074	0.147
BH (X_2_)	0.326	0.088	0.239	0.011		0.025	0.014	0.004	0.025	0.085	0.075
CC (X_3_)	0.633	0.074	0.559	0.035	0.03		0.042	0.039	0.036	0.144	0.233
PC (X_4_)	0.512	0.064	0.449	0.025	0.019	0.049		0.043	0.027	0.121	0.165
CD (X_5_)	0.427	0.087	0.34	0.024	0.004	0.034	0.031		0.024	0.081	0.142
CW (X_6_)	0.571	0.079	0.493	0.018	0.028	0.034	0.022	0.027		0.156	0.208
HB (X_7_)	0.681	0.248	0.432	0.021	0.03	0.043	0.031	0.028	0.05		0.229
BW (X_8_)	0.734	0.433	0.301	0.024	0.015	0.04	0.024	0.029	0.038	0.131	

**Table 4 animals-12-01886-t004:** Multiple linear regression analysis results.

Traits	Coefficients	Std. Error	T-Value	*p*	Tol	VIF
(Constant)	−386.562	104.548	−3.697	0.000		
BSL	2.899	1.862	1.557	0.121	0.738	1.355
BH	1.781	0.883	2.016	0.045	0.811	1.233
CC	1.830	1.571	1.165	0.246	0.379	2.637
PC	25.707	21.996	1.169	0.244	0.507	1.974
CD	2.889	1.578	1.831	0.069	0.684	1.462
CW	2.836	1.888	1.502	0.135	0.553	1.808
HB	10.290	2.402	4.284	0.000	0.458	2.184
BW	5.443	0.633	8.592	0.000	0.605	1.654

Note: R^2^ = 70.7%; R^2^ (adjusted) = 69.4%.

**Table 5 animals-12-01886-t005:** Factor analysis results.

	Factor Score Coefficients		Rotated Factor Loadings and Communalities		
Variables	Factor1	Factor2	Factor1	Factor2	Communality
BSL	0.561	−0.357	0.654	0.124	0.443
BH	0.419	0.628	−0.124	0.745	0.570
CC	0.833	−0.113	0.684	0.488	0.706
PC	0.715	−0.285	0.716	0.282	0.592
CD	0.585	−0.535	0.793	0.010	0.629
CW	0.702	0.333	0.284	0.723	0.604
HB	0.796	0.257	0.404	0.732	0.699
BW	0.741	0.043	0.511	0.539	0.552
Variance			4.464	1.089	5.553
Variance%			49.596	12.095	61.691

**Table 6 animals-12-01886-t006:** Regression analysis results according to the factor analysis results.

Traits	Coefficients	Std. Error	T-Value	*p*	Tol	VIF
(Constant)	566.365	1.154	490.613	<0.001		
Factor1	18.816	1.157	16.259	<0.001	1.0	1.0
Factor2	21.730	1.157	18.777	<0.001	1.0	1.0

Note: R^2^ = 75.8%; R^2^ (adjusted) = 75.6%.

## Data Availability

All data generated or analyzed during this study are included in this published article.

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
