# Peer review of "Stepwise Method and Factor Scoring in Multiple Regression Analysis of Cashmere Production in Liaoning Cashmere Goats"

_animals, 2022, doi:10.3390/ani12151886_

Round 1
Reviewer 1 Report
The paper describes relationships between cashmere wool production and body measures for 200 goats. This is done using several methods for describing linear relationships.
There are quite a few typographical errors in the manuscript : missing spaces etc. Some from the first part of the manuscript are shown below.
The explanation of forward and backward regression should be clarified. Someone not familiar with the methods will probably not understand what you did. Better use of English words is needed.
The relation of the significance level and the F test statistic is not clear.
I don't think you mean binary regression, but a model with two explanatory variables.
How the factor analysis was done is poorly described.
Path analysis is not described as a method.
What do you mean by 'Standard regression equation' ?
Simply Summary: This study investigated the relationship between certain body measurements
->
Simple Summary: This study investigated the relationship between certain body measurements
farms in Benxi County, Liaoning Province(East longitude 123°34’~125°46’, north latitude
->
farms in Benxi County, Liaoning Province (East longitude 123°34’~125°46’, north latitude
was designed according to the NRC [12] standard, and the vaccines were regularly vac-
->
was designed according to the NRC [12] standard. They were regularly vac-
cinated and dewormed on time.After the measurement and collection, the standard
->
cinated and dewormed on time. After the measurement and collection, the standard
Y changes uniformly with the change in the independent variable ??when other inde- -> X should not be italic :
->
Y changes uniformly with the change in the independent variable X? when other inde-
In the equation, β0, β1, β2,..,βk are regression parameters, andϵ is an unmeasurable
->
In the equation, β0, β1, β2,..,βk are regression parameters, and ϵ is an unmeasurable
tion of ϵ has zero mean, E(ϵ)=0. The variance of ϵ does not change as X?? chang- -> What does X?? mean ? The double subscript has unclear meaning.
es, D(ϵ)=σ2. ϵ has no autocorrelation,?ℴ?(ϵ?,ϵ?)=0. ϵ is uncorrelated with either ex-
planatory variable ??,?ℴ?(ϵ?,X?)=0. There is no perfect collinearity between the ex-
->
es, D(ϵ)=σ2. ϵ has no autocorrelation, ?ℴ?(ϵ?,ϵ?)=0. ϵ is uncorrelated with either ex-
planatory variable ??, ?ℴ?(ϵ?,X?)=0. There is no perfect collinearity between the ex-
Further typos are not shown. A thorough correction is needed !
Reviewer 2 Report
Introduction:
The introduction is very brief and do not provide any relevant information as regards Liaoning cashmere goats and factors relevant for cashmere production. The focus is too much on stressing the statistical analyses that goes beyond the scope of the journal.
Material and methods
The methods are not satisfactorily described to justify the scientific soundness of the study. I miss more precise information on the design of the experiment, animal status, feeding practices, live condition, selection criteria of goats and other specific factors that could potentially impact the results.
What software did the Authors use for all the statistical analyses?
Results
The presentation of results should clearly highlight aspects relevant for the study cashmere production instead of going into so many details on data analysis. The approach to data analysis should be summarized in a comprehensive manner within material and methods.
Discussion
The discussion does not sufficiently address the issues related to factors related to cashmere production.
Conclusions
There are no specific conclusions of the study rather a short summary of what was presented in the result and discussion. What your study adds to existing findings on cashmere production and goat breeding and selection? I am not truly convinced that the findings bring any new insights. Do the results have any implications for the producers? Did Authors identify any limitations of the study?
Round 2
Reviewer 1 Report
My suggested typographical changes from my first review do not seem to have been adressed :
Simply Summary -> Simple Summary
Space before ( in :
farms in Benxi County, Liaoning Province(East longitude 123°34’~125°46’, north latitude
->
farms in Benxi County, Liaoning Province (East longitude 123°34’~125°46’, north latitude
I only give the two first. Authors say they have revised typography, but I can't see it in the two examples above.
Author Response
Response to Reviewer 1 Comments
Point 1: My suggested typographical changes from my first review do not seem to have been adressed.
Response 1:Sorry, the previous modification of the format may not be reflected due to the software version. In the re-uploaded manuscript, I have modified the format problems of the article.
Reviewer 2 Report
Dear Authors,
The paper has been improved but there is a room for some improvements as regards the issues raised in my previous review. The presentation of results should focus more on aspects relevant for the audience of Animals and affecting the cashmere yield instead of going into so many details on data analysis.
Author Response
Response to Reviewer 2 Comments
Point 1: The presentation of results should focus more on aspects relevant for the audience of Animals and affecting the cashmere yield instead of going into so many details on data analysis.
Response 1:I have deleted the description of unnecessary data analysis results in the summary, highlighting the purpose and significance of this paper.
Round 3
Reviewer 2 Report
Dear Authors,
The paper has been improved but there are still some corrections needed as regards:
- Abstract - I do not see a point in quoting the regression equation in the abstract. Focus more on the implication of your study.
-The introduction is much too reduced and as such does not provide much relevant information.
- The discussion should highlight more the aspects essential for breeding and affecting the cashmere yield production.
- Conclusions – there is a need to add limitations and some recommendation for studies of similar nature.
Author Response
Point 1: Abstract - I do not see a point in quoting the regression equation in the abstract. Focus more on the implication of your study.
Response 1:The formula has been removed and research implications have been added
Point 2: The introduction is much too reduced and as such does not provide much relevant information.
Response 2:Introduction has been added.
Point 3: The discussion should highlight more the aspects essential for breeding and affecting the cashmere yield production.
Response 3:The discussion section has also been changed as you suggested.
Point 4: Conclusions – there is a need to add limitations and some recommendation for studies of similar nature.
Response 4:Added restrictions and recommendations based on your suggestion.